# A Theory of Tournament Representations

**Arun Rajkumar**
Indian Institute of Technology
RBCDSAI, IITM

**Abdul Bakey Mir**
Indian Institute of Technology

**Vishnu Veerathu**
Cohesity Inc

## Abstract

Real world tournaments are almost always intransitive. Recent works have noted that parametric models which assume $d$ dimensional node representations can effectively model intransitive tournaments (Rajkumar & Agarwal (2016)). However, nothing is known about the structure of the class of tournaments that arise out of any fixed $d$ dimensional representations. In this work, we develop a novel theory for understanding parametric tournament representations. Our first contribution is to structurally characterize the class of tournaments that arise out of $d$ dimensional representations. We do this by showing that these tournament classes have forbidden configurations which must necessarily be union of flip classes, a novel way to partition the set of all tournaments. We further characterize rank 2 tournaments completely by showing that the associated forbidden flip class contains just 2 tournaments. Specifically, we show that the rank 2 tournaments are equivalent to locally-transitive tournaments. This insight allows us to show that the minimum feedback arc set problem on this tournament class can be solved using the standard Quicksort procedure. For a general rank $d$ tournament class, we show that the flip class associated with a coned-doubly regular tournament of size $\mathcal{O}(\sqrt{d})$ must be a forbidden configuration. To answer a dual question, using a celebrated result of Forster & Simon (2006), we show a lower bound of $\Omega(\sqrt{n})$ on the minimum dimension needed to represent all tournaments on $n$ nodes. For any given tournament, we show a novel upper bound on the smallest representation dimension that depends on the least size of the number of unique nodes in any feedback arc set of the flip class associated with a tournament. We show how our results also shed light on upper bound of sign-rank of matrices.

## 1 Introduction

In this work, we lay the the foundations for a theory of tournament representations. A tournament is a complete directed graph and arises naturally in several applications including ranking from pairwise preferences, sports modeling, social choice, etc. We say that a tournament $\mathbf{T}$ on $n$ nodes can be *represented* in $d$ dimensions if there exists a skew symmetric matrix $\mathbf{M} \in \mathbb{R}^{n \times n}$ of rank $d$ such that a directed edge from $i$ to $j$ is present in $\mathbf{T}$ if and only if $M_{ij} > 0$. Real world tournaments are almost always intransitive (Tversky (1969); Klimenko (2015)) and it is not known what type of tournaments can be represented in how many dimensions. This is important to understand because of the following reason: As a modeler of preference relations using tournaments, it is often more natural to have *structural* domain knowledge such as *'The tournaments under consideration do not have long cycles'* as opposed to *algebraic* domain knowledge such as *'The rank of the skew symmetric matrix associated with the tournaments of interest is at most $k$'*. However, algorithms that learn rankings from pairwise comparison data typically need as input the algebraic quantity - the rank of the skew symmetric matrices associated with tournaments or equivalently the dimension where they are represented (Rajkumar & Agarwal (2016)). To bridge the gap between the structural and the algebraic world, we ask and answer two fundamental questions regarding the representations of tournaments.

*1) What structurally characterizes the class of tournaments that can be represented in $d$ dimensions?*

*2) Given a tournament $\mathbf{T}$ on $n$ nodes, what is the minimum dimension $d$ needed to represent it?.*

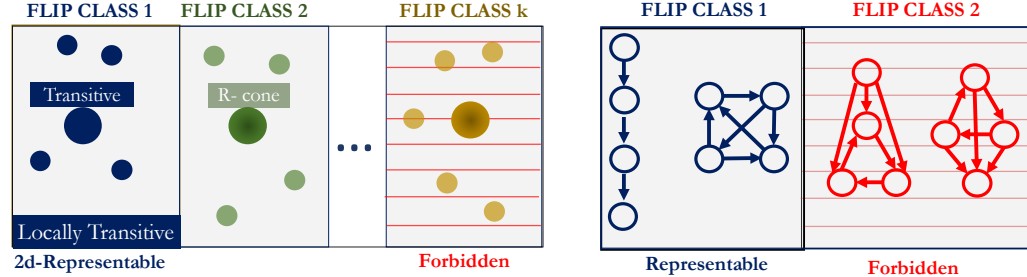

Figure 1: **(Left)** Partitions of the set of all tournaments on $n$ nodes using flip classes. Every shaded region is a flip class partition and every circle indicates a tournament. The flip class that contains the transitive tournament (Flip class 1) is precisely the set of all *locally transitive tournaments*. This is also the set of all tournaments that can be represented in 2 dimensions (Section 5). Every flip class contains a canonical representative termed the R-cone (Section 4), indicated using the larger circle inside each flip class. The tournaments that cannot be represented using $d$ dimensions appear as union of Forbidden Flip classes (Flip class $k$ in Figure) (Section 4). **(Right)** Explicit flip class partition of the 4 possible non-isomorphic tournaments on 4 nodes. Tournaments in flip class 1 can be represented using 2 dimensions whereas tournaments in flip class 2 cannot (see Section 5).
.

We answer the first question by investigating the intricate structure of the rank $d$ tournament class via the notion of forbidden configurations. Specifically, we show that the set of forbidden configuration for the rank $d$ tournament class must necessarily be a union of *flip classes*, a novel way to partition the set of all tournaments into equivalence classes. We explicitly characterize the forbidden configurations for the rank 2 tournament class and exhibit a forbidden configuration for the general rank $d$ tournament class. Specifically, we show that the rank 2 tournaments are equivalent to *locally transitive tournaments*, a previously studied class of tournaments (Cohen et al. (2004)). Our results throw light on the connections between transitive and locally transitive tournaments and also lets us develop a classic Quicksort based algorithm to solve that the minimum feedback arc set problem on rank 2 tournaments with $\mathcal{O}(n^2)$ time complexity. Our results for the general rank $d$ tournament class have connections to the classic long standing Hadamard conjecture and we discuss this as well. Figure 1 gives a glimpse of some of the main results.

We answer the second question by proving lower and upper bounds on the smallest dimension needed to represent a tournament on $n$ nodes. We exhibit a lower bound of $\Omega(\sqrt{n})$ using a variation of the celebrated dimension complexity result of Forster & Simon (2006) for sign matrices. To show upper bounds, we introduce a novel parameter associated to a tournament called the *Flip Feedback Node set of a Tournament*. This quantity depends on the least number of unique nodes in any feedback arc set of an associated tournament class for the tournament of interest and upper bounds linearly the representation dimension of any tournament. We show how our results can be used to provide upper bounds on the classic notion of sign-rank of a matrix. Previously known upper bounds for sign rank depended on the VC dimension of the associated binary function class (Alon et al. (2016)). Our upper bounds on the other hand have a graph theoretic flavour.

**Organization of the Paper:** We discuss briefly in Section 2 the foundational works this paper builds upon. We introduce necessary preliminaries in Section 3. The answer to the first question about the structural characterization of $d$ dimensional tournament classes span Sections 4, 5 and 6. We devote Section 7 to answer the second question about the number of dimensions needed to represent a tournament. Section 8.1 explores connections of our results to upper bounds on the sign rank of a sign matrix. Finally, we conclude in Section 10. All proofs are defered to the Appendix.

## 2  RELATED WORK

The work in this paper builds on several pieces of work across different domains. We summarize below the most important related works under different categories:

**Intransitive Pairwise Preference Models:** One of the main reasons to study representations of tournaments is to model pairwise preferences. Parametric pairwise preference models that can model intransitivity have gained recent interest. Rajkumar & Agarwal (2016) develop a low rank pairwise ranking model that can model intransitivity. However, their study and results were restricted to just the transitive tournaments in these classes. A generalization of the classical Bradley-Terry-Luce model (Bradley & Terry (1952), Luce (2012)) was studied in Causeur & Husson (2005). However, no structural characterization is known. Same holds for the more recent models studied in Chen & Joachims (2016), Makhijani & Ugander (2019) and Bower & Balzano (2020).

**Flip Classes:** The notion of flip classes, a novel way to partition the set of all tournaments on $n$ nodes, was first introduced in Fisher & Ryan (1995). The goal however was completely different and was on studying equilibrium on certain generalized rock-paper-scissors games on tournaments. Interestingly, and perhaps surprisingly, the notion of flip classes turn out to be fundamental to our study of understanding forbidden configurations of $d$ dimensional tournament classes.

**Dimension Complexity and Sign Rank:** Dimension complexity and sign rank of sign pattern matrices were studied in Forster (2002). These results have found significant applications in learning theory and lower bounds in computational complexity (Forster & Simon (2006)). More recently, Alon et al. (2016) study the sign rank for function classes with fixed Vapnik-Chervonenkis (VC) dimension and show upper bounds. Our upper bounds however depend on certain graph theoretic properties.

## 3 PRELIMINARIES

**Tournaments:** A tournament $\mathbf{T}$ is a complete directed graph i.e., a graph on $n$ nodes where for every pair of nodes $(i, j)$ either an edge is oriented from $i$ to $j$ or $j$ to $i$. The number of nodes $n$ in $\mathbf{T}$ will be usually clear from the context or will be explicitly specified. For nodes $i$ and $j$ in $\mathbf{T}$, we say $i \succ_{\mathbf{T}} j$ if there is a directed edge from $i$ to $j$. Given a node $i$, we define the out and in neighbours of $i$ as $\mathbf{T}_i^+ = \{j : i \succ_{\mathbf{T}} j\}$ and $\mathbf{T}_i^- = \{j : j \succ_{\mathbf{T}} i\}$ respectively. Given a set of nodes $S$, we denote by $\mathbf{T}(S)$ the induced sub-tournament of $\mathbf{T}$ on the nodes in $S$.

**Feedback Arc Set and Pairwise Disagreement Error:** Given a permutation $\sigma$ on $n$ nodes, the feedback arc set of $\sigma$ w.r.t the tournament $\mathbf{T}$ is defined as $E^\sigma(\mathbf{T}) = \{(i, j) : \sigma(i) > \sigma(j), i >_{\mathbf{T}} j\}$. The pairwise diagreement error of $\sigma$ w.r.t $\mathbf{T}$ is given by $\frac{|E^\sigma(\mathbf{T})|}{\binom{n}{2}}$. It is known that finding the $\sigma$ that minimizes the pairwise disagreement error w.r.t a general tournament $\mathbf{T}$ is a NP-hard problem (Charbit et al. (2007)).

**Skew Symmetric Tournament Classes:** A square matrix $\mathbf{M} \in \mathbf{R}^{n \times n}$ is skew symmetric if $M_{ij} = -M_{ji} \forall i, j$. In this paper, whenever we refer to a skew symmetric matrix $\mathbf{M} \in \mathbb{R}^{n \times n}$, we always assume $M_{ij} \neq 0 \quad \forall i \neq j$ and $M_{ii} = 0 \; \forall i$. Given such an $\mathbf{M}$, we denote by $\mathbf{T}\{\mathbf{M}\}$ the tournament on $n$ nodes *induced* by $\mathbf{M}$ where $i \succ_{\mathbf{T}} j \iff M_{ij} > 0$. We refer to class of tournaments induced by rank $d$ skew symmetric matrices as the *rank $d$ tournament class*.

**Forbidden Configurations:** A tournament class $\mathcal{T}$ is a collection of tournaments. $\mathcal{T}$ is said to *forbid* a tournament $\mathbf{T}$ if no tournament in $\mathcal{T}$ has a sub-tournament that is isomorphic to $\mathbf{T}$. We call $\mathbf{T}$ a *forbidden configuration* for $\mathcal{T}$ if $\mathcal{T}$ forbids $\mathbf{T}$ but does not forbid any sub-tournament of $\mathbf{T}$. For example, the class of all acyclic/transitive tournaments has the 3-cycle as a forbidden configuration i.e, the tournament $\mathbf{T}$ on three nodes $i, j, k$ where $i \succ_{\mathbf{T}} j \succ_{\mathbf{T}} k \succ_{\mathbf{T}} i$.

**Representations and Tournaments:** The matrix $\mathbf{A}^{\mathtt{rot}} \in \mathbb{R}^{d \times d}$ is defined for every even $d$ and is a block diagonal matrix which consists of $d/2$ blocks of $[0 - 1; 1\, 0]$. This is the canonical representation of a non-degenerate skew symmetric matrix i.e., any non-singular skew symmetric matrix can be brought to this form by a suitable basis transformation. Given a set of vectors in $\mathbf{H} = \{\mathbf{h}_1, \ldots, \mathbf{h}_n\}$ where each $\mathbf{h}_i \in \mathbb{R}^d$ for some even $d$, we refer to the set $\mathbf{H}$ as a *tournament inducing representation* if $\mathbf{h}_i^T \mathbf{A}^{\mathtt{rot}} \mathbf{h}_j \neq 0$ for all $i \neq j$. Furthermore, we refer to the tournament induced by $\mathbf{H}$ as $\mathbf{T}[\mathbf{H}]$ where a directed edge exits from node $i$ to $j$ if and only if $\mathbf{h}_i^T \mathbf{A}^{\mathtt{rot}} \mathbf{h}_j > 0$. It is easy to verify that $\mathbf{h}^T \mathbf{A}^{\mathtt{rot}} \mathbf{h} = 0$ for any $\mathbf{h} \in \mathbb{R}^d$. Any skew symmetric matrix $\mathbf{M} \in \mathbb{R}^{n \times n}$ of rank $d$ can be written as $\mathbf{M} = \mathbf{H}^T \mathbf{A}^{\mathtt{rot}} \mathbf{H}$ for some $\mathbf{H} \in \mathbb{R}^{d \times n}$. It follows that if $\mathbf{M} = \mathbf{H}^T \mathbf{A}^{\mathtt{rot}} \mathbf{H}$, then $\mathbf{T}\{\mathbf{M}\} = \mathbf{T}[\mathbf{H}]$.

**Positive Spans:** A set of vectors $\mathbf{H} = \{\mathbf{h}_1, \ldots, \mathbf{h}_n\}$ where each $\mathbf{h}_i \in \mathbb{R}^d$ is said to *positively span* $\mathbb{R}^d$ if for any $\mathbf{w} \in \mathbb{R}^d$, there exists non-negative constants $c_1, \ldots, c_n \geq 0$ such that $\sum_i c_i \mathbf{h}_i = \mathbf{w}$. If

$\mathbf{H}$ positively spans $\mathbb{R}^d$, then there does not exist a $\mathbf{w} \in \mathbb{R}^d$ such that $\mathbf{w}^T \mathbf{h}_i > 0 \ \forall i$. This is a easy consequence of Farkas' Lemma.

**Remark on Notation:** We reiterate that we use $\mathbf{T}(\cdot), \mathbf{T}\{\cdot\}$ and $\mathbf{T}[\cdot]$ to mean different objects - the tournament induced by a subset of nodes, the tournament induced by a skew symmetric matrix and the tournament induced by a representation of a set of vectors respectively. These will be usually clear from the context.

## 4   FLIP CLASSES, FORBIDDEN CONFIGURATIONS AND POSITIVE SPANS

The main purpose of this section is understand the space of forbidden configurations of rank $d$ tournaments. The main result of this section shows that the forbidden configurations for rank $d$ tournament classes occur as union of certain carefully defined equivalence classes of non-isomorphic tournaments. Towards this, we define the notion of *flip classes* which was introduced first in (Fisher & Ryan, 1995) although in a different context:

**Definition 1.** *Given a tournament $\mathbf{T}$ on $n$ nodes and a set $S \subseteq [n]$, define $\phi_S(\mathbf{T})$ to be the tournament obtained from $\mathbf{T}$ by reversing the orientation of all edges $(i, j)$ such that $i \in S, j \in \bar{S}$.*

In other words, $\phi_S(\mathbf{T})$ is obtained from $\mathbf{T}$ by reversing the edges across the cut $(S, \bar{S}) = \{(i, j) : i \in S, j \in \bar{S}\}$.

**Definition 2.** *A class of tournaments on $n$ nodes is called* cut-equivalent *if for every pair of tournaments $\mathbf{T}, \mathbf{T}'$ in the class, there exists a $S \subseteq [n]$ such that $\mathbf{T}'$ is isomorphic to $\phi_S(\mathbf{T})$*

It is easy to show that the set of cut-equivalent tournaments form an equivalence relation over the set of all tournaments on $n$ nodes Fisher & Ryan (1995). The corresponding equivalence classes are called a flip classes. We denote by $\mathcal{F}(\mathbf{T})$ the *flip class of* $\mathbf{T}$ i.e., the equivalence class of all cut-equivalent tournaments to $\mathbf{T}$. In the following theorem we show the fundamental relation between flip classes and forbidden configurations.

**Theorem 1.** *Let $\mathbf{T}$, a tournament on $k$ nodes, be a forbidden configuration for some rank $d$ tournament class. Then every tournament in the flip class of $\mathbf{T}$ is also a forbidden configuration for the rank $d$ tournament class.*

**Corollary 1.** *(Structure of Forbidden configurations) Forbidden configurations of rank $d$ tournament classes are unions of flip classes.*

Thus, to characterize the forbidden configuration of rank $d$ tournament classes, we need to understand the flip classes. We begin with the following simple but useful definition.

**Definition 3.** *A tournament $\mathbf{T}$ is called a $\mathbf{R}$-cone if there exists a node $i^*$ such that*

- $i^* \succ_{\mathbf{T}} j$ *for all $j \neq i^*$* $\left( j \succ_{\mathbf{T}} i^* \text{ for all } j \neq i^* \right)$

- $\mathbf{T}(\mathbf{T}_{i^*}^+) \left( \mathbf{T}(\mathbf{T}_{i^*}^-) \right)$ *is isomorphic to the tournament $\mathbf{R}$.*

$\mathbf{R}$-cones are essentially the tournament $\mathbf{R}$ along with an additional node that either beats or loses to nodes in $\mathbf{R}$. $\mathbf{R}$-cones are useful as they be viewed in some sense as canonical tournaments in flip classes. This is justified because of the following observation.

**Proposition 1.** *Every flip class contains an $\mathbf{R}$-cone for some tournament $\mathbf{R}$.*

The above observation says that to identify the forbidden configurations for a given tournament class, it suffices to identify all forbidden $\mathbf{R}$-cones. Then by Corollary 1, the associated flip classes will be the set of all forbidden configurations. However, it does not throw light on what property the tournament $\mathbf{R}$ must satisfy. The following lemma establishes this.

**Lemma 2.** *Let $\mathbf{R}$ be a tournament with the property that if $\mathbf{R} = \mathbf{R}[\mathbf{H}]$ for some representation $\mathbf{H} = \{\mathbf{h}_1, \ldots, \mathbf{h}_n\} \in \mathbb{R}^d$ then $\mathbf{H}$ positively spans $\mathbb{R}^d$. Then rank $d$ tournament class forbids $\mathbf{R}$-cones.*

The above lemma is extremely helpful in the sense that it reduces the study of forbidden configurations to the study of finding tournaments such that any representation that induces it must necessarily

positively span the entire Euclidean space. Note that there may be several representations which positively span the entire space. This does not mean their the associated coned tournaments are forbidden configurations. Instead, we start with an $\mathbf{R}$ cone and conclude it is forbidden if *every* representation that induces $\mathbf{R}$ necessarily positively spans the entire space. It is a non-trivial problem to identify such tournaments $\mathbf{R}$ for an arbitrary dimension $d$.

In the following sections, we explicitly identify the only forbidden flip class for rank 2 tournaments, one forbidden flip class for rank 4 and then (a potentially weaker) forbidden flip class for the general rank $d$ case.

## 5 Rank 2 Tournaments $\iff$ Locally Transitive Tournaments

The goal of this section is to characterize the forbidden configurations of rank 2 tournaments. Thanks to Lemma 2, this reduces to the problem of identifying a tournament whose representation necessarily span the entire space. The following lemma exhibits this tournament.

**Lemma 3.** *Let $\mathbf{H} = \{\mathbf{h}_1, \mathbf{h}_2, \mathbf{h}_3\} \in \mathbb{R}^2$ be two dimensional representation of 3 nodes which induce a 3 cycle tournament. Then the set $\mathbf{H}$ positively spans $\mathbb{R}^2$. Furthermore, the 3 cycle is the only such tournament on 3 nodes.*

The above lemma immediately implies that a coned 3-cycle is the only forbidden configuration for rank 2 matrices. This is indeed true. However, for the purposes of generalizing our result (which as we will see will be useful when discussing the higher dimension case), we will view the 3-cycle as a special case of a doubly regular tournament (defined next).

**Definition 4.** *A tournament $\mathbf{T}$ on $n$ nodes is said to be $n$-doubly regular if $|\mathbf{T}_i^+| = |\mathbf{T}_j^+|$ for all $i, j$ and $|\mathbf{T}_i^+ \cap \mathbf{T}_j^+| = k$ for some fixed $k$ for all $i \neq j$*

Trivially the 3-cycle is the only 3-doubly regular tournament. The following lemma establishes that the flip class of a coned 3 doubly regular tournament only contains itself.

**Proposition 2.** *The flip class of 3-doubly-regular-cone does not contain any other tournament.*

**Theorem 4.** *3-doubly-regular-cone is the only forbidden configuration for the rank 2 tournament class.*

We have thus far established that any rank 2 tournament on $n$ nodes forbids the 3-doubly-regular-cone. The advantage of this result is that we can go one step further and explicitly characterize the rank 2 tournament class. To do this, we need the definition of a previously studied tournament class.

**Definition 5.** *A tournament $\mathbf{T}$ is called* locally transitive *if for every node $i$ in $\mathbf{T}$, both $\mathbf{T}(\mathbf{T}_i^+)$ and $\mathbf{T}(\mathbf{T}_i^-)$ are transitive tournaments*

Before seeing why locally transitive tournaments are relevant to our study, we first show that they are intimately connected to transitive tournaments via the following characterization.

**Theorem 5.** *(**Connection between Transitive and Locally Transitive Tournaments**) The set of all non isomorphic locally transitive tournaments on $n$ nodes is equivalent to the flip class of the transitive tournament on $n$ nodes.*

We will see next that this key result allows us to immediately characterize the rank 2 tournament class. We will see later (Section 7) that this result is also crucial in determining an upper bound on the dimension needed to represent a given tournament.

**Theorem 6.** *(**Characterization of rank 2 tournaments**) A tournament $\mathbf{T}$ on $n$ nodes is locally transitive if and only if there exists a skew symmetric matric $\mathbf{M} \in \mathbb{R}^{n \times n}$ with rank($\mathbf{M}$) = 2 such that the $\mathbf{T}\{\mathbf{M}\} = \mathbf{T}$.*

It is perhaps surprising that a purely structural description of a tournament class namely that of local transitivity turns out to be exactly equivalent to the rank 2 tournament class. To the best of our knowledge, this characterization appears novel and hasn't been previously noticed. One of the interesting consequence of the above characterization is that the minimum feedback arc set problem on rank 2 tournaments can be solved using a standard quick sort procedure. This is formalized below.

**Theorem 7.** *(**Minimum Feedback Arc Set is Poly-time Solvable for Rank** 2 **tournaments**) Let* $\mathbf{T}$ *be a locally transitive tournament on $n$ nodes and let $\sigma_1$ be the permutation returned by running a standard quick sort algorithm choosing $1$ as the initial pivot node and where the outcome of comparison between items $i$ and $j$ is $i$ if and only if $i \succ_{\mathbf{T}} j$. Let $\sigma_k$ be obtained from $\sigma_1$ by $k - 1$ clockwise cyclic shifts for $k \in [n]$. Let $E_k$ be the feedback arc set of $\sigma_k$ w.r.t $\mathbf{T}$. Then $\min_k |E_k|$ achieves the minimum size of the feedback arc set for $\mathbf{T}$.*

*Proof.* (Sketch) The proof involves two steps: first arguing that by fixing any pivot, quick sort would return a ranking that is a cyclic shift of $\sigma_1$. The second step involves inductively arguing that one of the cyclic shift must necessarily minimize the feedback arc set. □

### 5.1 Rank 4 Tournaments

We now turn to rank $4$ tournaments. We could have directly considered rank $d$ tournaments, but it turns out that what we can show a slightly stronger result for rank $4$ tournaments than the general case and so we focus on them separately. While it is arguably simple in the rank $2$ case to identify the tournament that necessitates the positive spanning property, it is not immediately clear in the rank $4$ case. A first guess would be to consider the regular tournaments (as the $3$ cycle for rank $2$ is also a regular tournament) on $5$ nodes or $7$ nodes. However, these turn out to be insufficient as one can construct counter examples of regular tournaments on up to $7$ nodes with representations that don't span the entire $\mathbb{R}^4$. In fact, as we had defined earlier, the right way to generalize to higher dimension turns out to be using doubly regular tournaments.

**Theorem 8.** *The $11$-doubly-regular-cone is a forbidden configuration for rank $4$ tournament class.*

Note that while for the rank $2$ case, we were able to prove that the *only* forbidden flip class is the one that contains a coned $3$ cycle, we have not shown that the only forbidden configuration for rank $4$ class is the $11$ doubly regular cone. In fact, we believe that the smallest forbidden tournament for rank $4$ class is the $7$- doubly regular cone. However, we haven't been able to prove this. This appears to be a non-trivial problem. From our simulation experiments, we observe that the *only* flip class that could be forbidden on $8$ nodes is the one that contains the $7$-doubly regular cone. In particular, we were able to produce examples of representations for all other flip classes on $8$ nodes. However, this does not imply that the ones where we could not produce a forbidden configuration is in fact forbidden. Unfortunately, it seems tricky to prove this and we don't have a way to show this at this point. On the other hand, as we will see in the next section, the result in Theorem 8 is still stronger than the result for the general rank $d$ tournaments.

Having discussed rank $2$ and rank $4$ cases separately, we next turn our attention to the general rank $d$ tournament class.

## 6 Rank $d$ tournaments and the Hadamard Conjecture

From the understanding of rank $2$ and rank $4$ tournament classes in the previous sections, and noting that the corresponding forbidden configurations are intimately related to doubly regular tournaments, it is tempting to conjecture that this is true in general.

**Conjecture 1.** *Rank $2d$ tournament class forbids $4d - 1$-doubly-regular-cones.*

Ideally, the conjecture above must have a qualifier 'if they exist' for the $4d - 1$ doubly regular cone. This is because of the equivalence between doubly regular tournaments on $4d - 1$ nodes and Hadamard matrices in $\{+1, -1\}^{4d \times 4d}$ (Reid & Brown (1972)). A matrix $\mathbf{H} \in \{+1, -1\}^{n \times n}$ is called Hadamard if $\mathbf{H}'\mathbf{H} = nI$ where $I$ is the identity matrix. It is known that there is a bijection between skew Hadamard matrices and doubly regular tournaments Reid & Brown (1972). A long standing unsolved conjecture about Hadamard matrices is the following:

**Conjecture 2.** *(**Hadamard**) There exists a Hadamard matrix of order $4d$ for every $d > 0$.*

If Conjecture 1 were true, then it would imply the existence of $4d - 1$-doubly regular tournament for every $d$ and thus would imply the Hadamard conjecture is true. In fact, Conjecture 1 being true would say more which we state below:

**Conjecture 3.** *There exists a skew symmetric Hadamard matrix of order $4d$ for every $d > 0$.*

The main result of this section is a weaker form of the conjecture:

**Theorem 9.** *Rank* $2(d-1)$ *tournament class forbids* $12d^2 - 1$*-doubly-regular-cones if they exist.*

## 7 How Many Dimensions are needed to Represent A Tournament?

The previous sections considered a specific rank $d$ tournament class and tried to characterize them using forbidden configurations. In this section, We turn to the dual question of understanding the minimum number of dimensions needed to embed a tournament. We start by not considering a single tournament $\mathbf{T}$ but the set of all tournaments on $n$ nodes. We show below a general result which provides a lower bound on the minimum dimension needed to embed any tournament on $n$ nodes.

**Theorem 10.** *(Lower Bound on minimum representation dimension) Let* $\mathbf{T}$ *be a tournament on* $n$ *nodes. Then there exists* $\mathbf{H} = \{h_1, \ldots, h_n\} \in \mathbb{R}^d$ *such that* $\mathbf{T} = \mathbf{T}[\mathbf{H}]$ *only if* $d \sum_{i=1}^{n} (\rho_i(\mathbf{T}+I))^2 \geq n^2$. *Furthermore, let* $d$ *be the minimum dimension needed to embed every tournament on* $n$ *nodes. Then,* $d \geq \sqrt{n}$.

The result follows the arguments in the celebrated work of Forster & Simon (2006). As Alon et al. (2016) point out, Forster's technique cannot be stretched further in obvious ways to get upper bounds.

The above theorem tells us that in the worst case at least $\Omega(\sqrt{n})$ dimensions are necessary to represent all tournaments on $n$ nodes. In fact if Conjecture 1 were true, the minimum dimension would be $\Omega(n)$. However, in practice one might not encounter tournaments with such extremal/worst-case properties.

**Remark:** The lower bound for the representation dimension of random tournaments (where the orientation of each edge is determined by an unbiased Bernoulli coin toss) will depend on the singular values of random tournaments. However, we don't expect the representation dimension to be of independent of $n$, the number of nodes. Loosely speaking, a doubly regular tournament is "like" a random tournament (the associated Hadamard Tournament has been used in deterministic perturbation schemes as alternatives for random sign matrices (Bhatnagar et al. (2003))). On the other hand, the most interesting real-world tournaments might be characterized by constant sized node representations and hence may be structurally much more constrained than random tournaments. Typically, a smaller number of dimensions might be enough to represent tournaments of practical interest.

Our goal below is to upper bound on the number of dimensions needed to embed a given tournament $\mathbf{T}$.

Recall that $E^\sigma(\mathbf{T})$ denotes the feedback arc set of a permutation $\sigma$ w.r.t a tournament $\mathbf{T}$. We define the number of nodes involved in the feedback arc set as follows: $\theta(\sigma, \mathbf{T}) = |\{i : \exists j : \sigma(i) > \sigma(j), (i,j) \in E^\sigma(\mathbf{T})\}|$. We next define a crucial quantity $\mu(\mathbf{T})$ which we term as the *Flip Feedback Node set size*. This quantity will determine an upper bound on the dimension where a tournament can be represented:

**Definition 6.** *Given a tournament* $\mathbf{T}$, *define the* Flip Feedback Node Set Size *as follows:*

$$\mu(\mathbf{T}) = \min_\sigma \min_{\mathbf{T}' \in \mathcal{F}(\mathbf{T})} \theta(\sigma, \mathbf{T}')$$

In words, given a tournament $\mathbf{T}$, the quantity $\mu(\mathbf{T})$ captures the minimum number of nodes involved in any feedback arc set among all tournaments in the flip class of $\mathbf{T}$. For instance if $\mathbf{T}$ is a locally transitive tournament then $\mu(\mathbf{T})$ would be $0$ - as $\mathbf{T}$ is necessarily in the flip class of a transitive tournament and so the $E^\sigma$ corresponding to the topological ordering of the transitive tournament would have an empty feedback arc set. As another example, consider $\mathbf{T}$ to be the coned 3-cycle. The flip class of this tournament contains only itself and the best permutation will have one edge in the feedback arc set. Thus $\mu(\mathbf{T}) = 1$. In general, it is trivially true that $\mu(\mathbf{T})$ is upper bounded by $n$, the number of nodes. However$\mu(\mathbf{T})$ could be much smaller than $n$ depending on $\mathbf{T}$. The main result of this section is the theorem below that shows that $\mu(\mathbf{T})$ gives an upper bound on the number of dimension needed to represent any tournament.

**Theorem 11.** *(Upper Bound on minimum representation dimension) For any tournament* $\mathbf{T}$, *there exists* $\mathbf{H} = \{\mathbf{h}_1, \ldots, \mathbf{h}_n\} \in \mathbb{R}^{2(\mu(\mathbf{T})+1)}$ *such that* $\mathbf{T} = \mathbf{T}[\mathbf{H}]$.

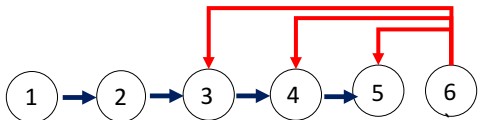

Figure 2: A simple tournament to illustrate that the quantity $\mu(\mathbf{T})$ need not be same as the size of the minimum feedback arc set. All edges go from left to right except the ones in red. See Remark 2 in Section 7 for details.

**Remark 1:** The above theorem says that one can always obtain an representation $\mathbf{H}$ in dimension $d = \mathcal{O}(\mu(\mathbf{T}))$ that induces $\mathbf{T}$. The bound gets tighter for tournaments with smaller feedback arc sets, which is what one might typically expect in practice. Note that even for some tournaments that may have a large feedback arc set, the associated flip class might contain a tournament with a smaller feedback arc set.

**Remark 2:** We note that in general $\mu(\mathbf{T})$ is *not necessarily* the cardinality of the minimum feedback arc set among all tournaments in the flip class of $\mathbf{T}$. Instead $\mu(\mathbf{T})$ captures the cardinality of the set of nodes involved in any feedback arc set. To see why these two could be different, consider the tournament in Figure 2. Here, $\sigma_a = [6\ 1\ 2\ 3\ 4\ 5]$ is the permutation minimizing the feedback arc set. Let $\sigma_b = [1\ 2\ 3\ 4\ 5\ 6]$. Note that $|E^{\sigma_a}| = 2$, $\theta(\sigma_a, \mathbf{T}) = |\{1, 2\}| = 2$. However $|E^{\sigma_b}| = 3$, yet $\theta(\sigma_b, \mathbf{T}) = |\{6\}| = 1$. Thus, $\sigma_b$ gives a tighter upper bound on the dimension needed to represent $\mathbf{T}$.

# 8 APPLICATIONS OF THE RESULTS

## 8.1 CONNECTIONS TO SIGN RANK

The sign rank of a matrix $\mathbf{G} \in \{+1, -1\}^{m \times n}$ is defined as the smallest integer $d$ such that there exists a matrix $M \in \mathbb{R}^{m \times n}$ of rank $d$ that satisfies $sign(M) = G$ [1]. Here $sign(z) = 1$ if $z > 0$ and $-1$ otherwise. A breakthrough result on the lower bound on the sign rank was given by Forster & Simon (2006). However, good upper bounds have been harder to obtain. We show below how Theorem 11 also translates to an upper bound on the sign rank of any sign matrix $G$.

**Theorem 12.** *Let $\mathbf{G} \in \{+1, -1\}^{m \times n}$ be an arbitrary sign matrix. Let $\mathbf{T}$ be any tournament on $m + n$ nodes. Let $S = \{1, \ldots, m\}$ and let edge orientations of the cut$(S, \bar{S})$ in $\mathbf{T}$ be determined by $\mathbf{G}$ Then sign-rank$(\mathbf{G}) \le 2(\mu(\mathbf{T}) + 1)$*

We argue that the the above result can give significantly tigher bounds than the bounds in Alon et al. (2016). The simplest example one can consider is a locally transitive tournament on $n$ nodes. Viewing this as a sign pattern matrix, it is easy to argue that the VC dimension of such a matrix is atmost $2$. Theorem $5$ in Alon et al. (2016) shows that the sign rank is upper bounded by $\mathcal{O}(\sqrt{n}))$ when VC dimension is $2$ (For a general hypothesis class of VC dimension $d$, the upper bound is $\mathcal{O}(n^{1-\frac{1}{d}})$. On the other hand, by definition $\mu(\mathbf{T})$ for any locally transitive tournament is exactly $0$ (and so the upper bound of $2\mu(\mathbf{T}) + 1$ is exactly $2$ and is tight) for any locally transitive tournament which is independent of the number of nodes $n$. The important reason for this significantly improved bound using $\mu(\mathbf{T})$ is that we consider only skew symmetric sign pattern matrices while Alon et al. (2016) look at the worst case (w.r.t representation dimension/sign rank) sign pattern matrices for a fixed VC dimension. Our bound reduces the study of sign rank to a more graph theoretic study of the feedback arc set problem. It is not clear if the upper bound of $2(\mu(\mathbf{T}) + 1)$ can be improved and we leave this to future work.

## 8.2 CONNECTIONS TO LEARNING FROM PAIRWISE COMPARISONS

Consider a learning to rank problem from pairwise comparisons. Here, a set of $n$ items need to be ranked from a subset of pairwise comparisons among them. Every pair that is chosen for comparison is compared a fixed number of times. Every time items $i$ and $j$ are compared, $i$ is preferred over $j$

---

[1]Sign-rank can be defined for any general matrix $G \in \mathbb{R}^{m \times n}$. We restrict to the sign matrices to make the connections to the tournament matrices explicit

with probability $\mathbf{P}_i j$. A common and popular model to capture these probabilities is the Bradley-Terry-Luce (BTL) model Bradley & Terry (1952) where $\mathbf{P}_{ij} = s_i/(s_i + s_j)$ for some score vector $\mathbf{s} \in \mathbf{R}^n$. Note that the model is completely specified by the score vector $\mathbf{s}$ which in turn completely determines the probability preference matrix $\mathbf{P}$. The learning problem is to learn these unknown score vector $\mathbf{s} \in R^n$ from noisy pairwise comparisons. Once the score vector is learnt, a ranking can be obtained by sorting these scores and the pairwise probabilities of unseen pairs of items can be predicted.

The above BTL model is an example of a rank-2 model in the sense that the probability preference matrix $\mathbf{P}$ results in a rank 2 matrix under the log-odds skew symmetric transformation. Indeed, if we define $M_{ij} = \log(\frac{P_{ij}}{P_{ji}})$. then equivalently $M_{ij} = \log(s_i) - \log(s_j)$. It is easy to show that $\mathbf{M}$ is a rank 2 skew symmetric matrix. However, the major disadvantage of the BTL model is that it can capture only transitive preferences i.e, $P_{ij} > 0.5$ and $P_{jk} > 0.5 \implies P_{ki} > 0.5$. In real world situations, intransitivity is very common. To achieve intransitivity, the simplest way would be to start with a general rank $r$ skew symmetric matrix $\mathbf{M}$ and consider the probability matrix that determines the preference probabilities as $P_{ij} = 1/(1 + \exp(-M_{ij}))$. Here, the learning problem would be to estimate the two score vectors or equivalently the two dimensional representation for each item. Previous studies (Rajkumar & Agarwal (2016)) show that this can be learnt using Matrix completion based approaches or maximum likelihood based approaches (Chen & Joachims (2016)). However, what was not known earlier is the structure of tournaments that can be captured using such low rank restrictions which is important to decide on the parameter $r$. Our work helps modellers make informed decision to choose $r$ based on explicitly identifying the types of tournaments that can be modelled and subsequently learnt.

We discuss next a simple application where modeling a ranking from pairwise comparisons problem can benefit from the insights gained using our results. Consider a situation of modelling sports tournaments such as Tennis. Here, one can choose to model the players (nodes) using 2-dimensions where the dimensions corresponds to their *offense* (forehand) and *defense* (backhand) strengths respectively. When two players compete, the advantage of the offense strength of player 1 w.r.t the defense of player 2 and vice versa determine the outcome of the match. This is precisely captured by a rank 2 model where the learning problem would be to infer these latent offense/defense strength of each player from outcomes of pairwise competitions. Theorem 6 says that such a model would immediately lead to locally transitive tournaments among the players. This structural characterisation now gives insights to the modeller if 2 dimensions are enough to model the players or not.

## 9 REAL WORLD TOURNAMENT EXPERIMENTS

We conducted simple experiments on real world data sets. Specifically, we considered 114 real world tournaments that arise in several applications including election candidate preferences, Sushi preferences, cars preferences, etc (source: www.preflib.org). The number of nodes in these tournaments varied from 5 to 23. Out of the 114 tournaments considered, 76(66.67%) were in fact locally transitive. For these tournaments, the upper bounds and lower bounds given by our theorems matched and was equal to 2. Interestingly, even for the non-locally transitive tournaments, the lower bound still turned out to be 2. We computed the upper bound for tournaments of size at most 9 (we did not do it for larger tournaments as this involves a brute force search) and found the value to be either 4 or 6. This shows that the upper bounds are usually non-trivial and efficiently approximating it is an interesting direction for future work.

## 10 CONCLUSION

In this work, we develop a theory of tournament representations. We show how fixing the representation dimension enforces, via forbidden configurations, restrictions on the type of tournaments that can be represented. We study and characterize rank 2 tournaments and show forbidden sub-tournaments for the rank $d$ tournament class. We develop upper and lower bounds for minimum dimension needed to represent a tournament. Future work includes attempting to look deeper into some of the conjectures presented and possible strengthening of some of the bounds presented.

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

## A    APPENDIX

**Proof of Theorem 1**

*Proof.* Assume there is a $\mathbf{T}' \in \mathcal{F}(\mathbf{T})$ which is not forbidden for rank $d$ tournament class. Thus, $\exists \mathbf{H}' = \{\mathbf{h}'_1, \dots, \mathbf{h}'_n\} \in \mathbb{R}^d$ such that $\mathbf{T}' = \mathbf{T}[\mathbf{H}']$. By definition, there must also exist a $S \subseteq [n]$ such that $\mathbf{T}' = \phi_S(\mathbf{T})$. Consider the representation $\mathbf{H}$ obtained from $\mathbf{H}'$ where $\mathbf{h}_i = -\mathbf{h}'_i$ for all $i \in S$ and $\mathbf{h}_i = \mathbf{h}'_i$ otherwise. It is easy to verify that $\mathbf{T} = \mathbf{T}[\mathbf{H}]$ which is a contradiction to the assumption that $\mathbf{T}$ is a forbidden configuration for rank $d$ tournaments. $\square$

**Proof of Proposition 1**

*Proof.* Consider any tournament $\mathbf{T}$. Let $\mathbf{T}' = \phi_{\{i \cup \mathbf{T}_i^-\}}(\mathbf{T})$ for an arbitrary node $i$. By definition $\mathbf{T}' \in \mathcal{F}(\mathbf{T})$. Also, $\mathbf{T}'$ is a $\mathbf{R}$-cone, coned by $i$. $\square$

**Proof of Lemma 2**

*Proof.* Consider any representation $\mathbf{H}$ that induces a tournament $\mathbf{R}$. By assumption of the theorem, $\mathbf{H}$ positively spans $\mathbb{R}^d$. Thus, by Farkas' lemma, there cannot exist a $\mathbf{v} \in \mathbb{R}^d$ such that $\mathbf{v}^T \mathbf{h}_i > 0 \ \forall i$. Note that if $\mathbf{R}$-cone is not a forbidden configuration for rank $d$ tournaments, then there must exist a $\mathbf{h} \in \mathbb{R}^d$ such that $\mathbf{h}^T A^{rot} \mathbf{h}_i > 0 \ \forall i$. As $A^{rot}$ is invertible, one can set $\mathbf{v} = (A^{rot})^T \mathbf{h}$ with the property that $\mathbf{v}^T \mathbf{h}_i > 0 \ \forall i$. But this contradicts the conclusion drawn earlier from Farka's lemma. $\square$

**Proof of Proposition 2**

*Proof.* This is easily verified by checking all tournaments in $\mathcal{F}(\mathbf{T})$ where $\mathbf{T}$ is the coned 3-cycle. $\square$

**Proof of Lemma 3**

*Proof.* Wlog, assume that $1 \geq_{\mathbf{T}} 2 \geq_{\mathbf{T}} 3 \geq_{\mathbf{T}} 1$. Then, it must be the case that the counterclockwise angle between the representation of the corresponding items must be $\leq 180$ degrees. However, if the representations $\{\mathbf{h}_1, \mathbf{h}_2, \mathbf{h}3\}$ did not positively span $\mathbf{R}^2$, then by Farka's Lemma, there must be some supporting hyperplane for the representations. However, this would imply at least one of the node pairs $\{(1,2), (2,3), (3,1)\}$ must necessarily make an angle $\geq 180$ degrees. But this contradicts the assumption that the nodes form a 3 cycle. $\square$

**Proof of Theorem 5**

*Proof.* Let $\mathbf{T}$ be a transitive tournament on $n$ nodes and let $\mathbf{T}' \in \mathcal{F}(\mathbf{T})$. Then, there exists some $S \subseteq [n]$ such that $\mathbf{T}' = \phi_S(\mathbf{T})$. Consider any node $i \in [n]$. Define following 4 subset of nodes associated with $i$: $S_1 = \mathbf{T}_i^+ \cap S, S_2 = \mathbf{T}_i^+ \backslash S_1, S_3 = \mathbf{T}_i^- \cap S, S_4 = \mathbf{T}_i^- \backslash S$. Note that each of $\mathbf{T}(S_k)$ for $k = 1$ to $4$ is a transitive sub-tournament and the relationship across $S_i, S_j$ for any two sets is either one completely beats the other or completely loses to the other. Note that in $\phi_S(\mathbf{T})$ the orientation of the edges across these sets is either flipped as a whole or not flipped at all. Thus, exactly two of these sets will be part of $\mathbf{T}_i'^+$ and two part of $\mathbf{T}_i'^-$ (the exact sets among these sets will depend on whether $i \in S$ or not), thus preserving the local transitivity property. Thus $\mathbf{T}'$ is locally transitive.

To prove the opposite direction, let $\mathbf{T}$ be a locally transitive tournament. Let $i \in [n]$ be an arbitrary node. We argue that $\mathbf{T}' := \phi_{\mathbf{T}_i^+}(\mathbf{T})$ is a transitive tournament. We will show this by arguing that there does not exist a 3 cycle in $\mathbf{T}'$. Consider any 3-cycle $a >_{\mathbf{T}} b >_{\mathbf{T}} c >_{\mathbf{T}} a$. As $\mathbf{T}$ is locally transitive, not all $\{a, b, c\}$ can be in $\mathbf{T}_i^+$. Also not all $\{a, b, c\}$ can avoid $\mathbf{T}_i^+$ as $[n] \setminus \mathbf{T}_i^+$ is transitive. Thus at least one and at most two of $\{a, b, c\}$ belongs to $\mathbf{T}_i^+(\mathbf{T})$. This means that the 3-cycle becomes a transitive tournament in $\mathbf{T}' := \phi_{\mathbf{T}_i^+}(\mathbf{T})$. Thus every cycle in $\mathbf{T}$ becomes transitive in $\mathbf{T}'$. Now consider any 3 nodes which forms a transitive tournament $a >_{\mathbf{T}} b >_{\mathbf{T}} c$ and involves at least one

node and at most two nodes in $\mathbf{T}_i^+$. Then there are only two cases to consider: (1) $a \in [n] \setminus \mathbf{T}_i^+$ and $\{b, c\} \in \mathbf{T}_i^+$ or (2) $\{a, b\} \in [n] \setminus \mathbf{T}_i^+$ and $c \in \mathbf{T}_i^+$. In both these cases, it is easy to verify that the the corresponding tournament in $\mathbf{T}'$ is either the transitive tournament $b >_{\mathbf{T}'} c >_{\mathbf{T}'} a$ or the transitive tournament $c >_{\mathbf{T}'} a >_{\mathbf{T}'} b$. As these are the only possibilities, the result follows by noting that $\mathbf{T}' = \phi_{\mathbf{T}_i^+}(\mathbf{T}) \implies \mathbf{T} = \phi_{\mathbf{T}_i^+}(\mathbf{T}')$ and so $\mathbf{T} \in \mathcal{F}(\mathbf{T}')$. □

**Proof of Theorem 6**

*Proof.* Assume $\mathbf{T}$ is locally transitive. Then by Theorem 5, it must be in the flip class of some transitive tournament $\mathbf{T}'$ i.e. $\mathbf{T} = \phi_S(\mathbf{T}')$ for some $S \subseteq [n]$. It is easy to represent a transitive tournament using a rank 2 skew symmetric matrix. Indeed pick any vector $\mathbf{u} \in \mathbb{R}^n$ which is sorted according to the topological ordering of $\mathbf{T}'$. Let $\mathbf{v} \in \mathbb{R}^n$ be the all ones vector. Then $\mathbf{M} = \mathbf{u}\mathbf{v}^T - \mathbf{v}\mathbf{u}^T$ represents $\mathbf{T}'$. Let $\mathbf{M} = (\mathbf{H}')^T A^{rot} \mathbf{H}$ for some $\mathbf{H}' \in \mathbb{R}^{2 \times n}$. Then the columns of $\mathbf{H}'$ represent $\mathbf{T}'$. Now consider the representation $\mathbf{H}'$ obtained from $\mathbf{H}'$ where the columns indexed by $S$ are multiplied by $-1$. This does not change the rank of $\mathbf{H}'$ and it can be verified that $\mathbf{T} = \mathbf{T}[\mathbf{H}]$.

To prove the other direction, consider any rank 2 skew symmetric matrix $\mathbf{M} \in \mathbb{R}^{n \times n}$. Then there must exist $\mathbf{u}, \mathbf{v} \in \mathbb{R}^n$ such that $\mathbf{M} = \mathbf{u}\mathbf{v}^T - \mathbf{v}\mathbf{u}^T$. Consider any node $i \in [n]$. Consider three nodes $a, b, c$ such that $u_i v_j > u_j v_i$ for $j \in \{a, b, c\}$. Furthermore let $u_a v_b > v_a u_b$ and $u_b v_c > v_b u_c$. Then by carefully going over all $\{\pm\}$ sign possibilities for $\{u_i, u_a, u_b, u_c, v_i, v_a, v_b, v_c\}$, one can conclude that it must be the case that $u_a v_c > v_a u_c$. This just shows that $\mathbf{T}_i^+$ is transitive. Analogously one can show that $\mathbf{T}_i^-$ is also transitive. As $i$ was arbitrary, the result follows. □

**Proof of Theorem 7**

*Proof.* Recall that the classic quick sort algorithm picks a pivot node (say 1) and places all nodes that *beat* the pivot to the right in the ranking and those that *lose* to the left and then recurses on the left and right subsets. As $\mathbf{T}$ is locally transitive, choosing any pivot $i$ would correspond to fixing the pivots position and simply returning the ranking $[\sigma(N_i^-), i, \sigma(N_i^+)]$ where $\sigma(N_i^+), \sigma(N_i^-)$ correspond to the topological ordering of the transitive tournaments $\mathbf{T}(N_i^+)$ and $\mathbf{T}(N_i^-)$ respectively. We first argue that changing the pivot only cyclically shifts the final ranking. To see why this is true, consider two pivots $i$ and $j$ and their corresponding rankings $\sigma^i$ and $\sigma^j$. Without loss of generality, assume that the ranking $\sigma^i = [1, \ldots, n]$ and $i >_{\mathbf{T}} j$. We argue that there exists an integer $k$ such that $N_j^+ = \{j + 1 \mod n, (j + 2) \mod n, \ldots, (j + k) \mod n\}$ (where by convention $n \mod n = n$). As $N_i^+$ is transitive, and $j$ is part of it, it must be the case that all the nodes $\{j + 1, \ldots, j + n\} \in N_j^+$. Then to prove the claim, it remains to be shown that the set $N_i^- \cap N_j^+$ is either empty or must be the nodes $\{1, \ldots, \ell\}$ for some $\ell < i$. If empty, we are done. If not, assume for the sake of contradiction that there exists three successive integers $\ell_a, \ell_b, \ell_c < i$ such that $\ell_a, \ell_c <_{\mathbf{T}} j$ but $\ell_b >_{\mathbf{T}} j$. It is easy to verify that this cannot happen as it would lead to $\mathbf{T}(\{\ell_a, \ell_b, \ell_c, j\})$ being a forbidden configuration.

Notice that as every locally transitive tournament is in the flip class of a transitive tournament i.e., $\mathbf{T} = \phi_S(\mathbf{T})$ for some $S$, one can divide the set of all nodes into $2k + 1$ groups as follows: Let $[1, \ldots, n]$ be the ordering corresponding to the transitive tournament wlog. Starting from 1, add as many nodes to a group such that all the elements belong to either $S$ or $\bar{S}$. Once the condition is violated, create a new group and continue the same process. It is not hard to verify that $2k + 1$ groups will be formed in this process for some $k \geq 0$. Moreover, each group would *separate* two other groups by construction.

We can first show that the items of a single group must appear in consecutive positions in one of the optimal rankings. This is proven as follows.

Consider there exists an optimal ranking with items which belong to the same group not occurring consecutively. Consider two items belonging to the same group, which have items from other groups present in between them in the ranking. Consider these items to be $a_1, a_2$, with $a_1$ present above in the rankings. Consider the number of upsets that the two items are involved in to be $u_1$ and $u_2$. If $u_1 \leq u_2$, $a_2$ can be placed right after $a_1$ in the ranking, creating a better or equivalent ranking in terms of upsets. Similarly if $u_1 \geq u_2$, $a_1$ can be placed directly above $a_2$ in the rankings to create

an equivalent or better ranking. Therefore there exists an optimal ranking which has all items in the same group consecutively.

This theorem is then reduced to finding a ranking of groups, which is proven using induction on $k$.

**Base Case**
Consider the base case with $k = 1$. Let there be 3 groups, $C$, $A_1$, $B_1$. We can say that the optimal ranking cannot be any of the following

$$CB_1A_1$$

$$A_1CB_1$$

$$B_1A_1C$$

since all three rankings can be made better by swapping the second and third ranked groups. Therefore the 3 possible optimal rankings are

$$CA_1B_1$$

$$A_1B_1C$$

$$B_1CA_1$$

which are cyclic shifts of each other.

**Inductive Step**
One property of rankings which is useful for the inductive step proof is as follows. Let there be $2k + 1$ groups $G = \{g_1, g_2 \dots g_{2k+1}\}$. Label the optimal ranking with the condition that $g_i$ be placed first in the ranking as $R_i$. The ranking $R_i$ with $g_i$ removed must be the optimal ranking for $G \setminus \{g_i\}$. This can be shown using contradiction i.e, if there was a better ranking for $G \setminus g_i$, that ranking with $g_i$ appended to the front would be better than $R_i$.

We now assume the theorem is true for size $2k - 1$ instances and aim to prove for the same for size $2k + 1$ instances. Consider $g_1$ as the first group in the ranking. This creates a certain number of upsets, for the purposes of ranking the remaining groups, 2 of the remaining groups can be merged into a single group. This follows from the observation earlier that each group also 'separates' two groups. This can be considered an instance of the size $2k - 1$ problem. Therefore the set of optimal rankings with $g_1$ as the first group in the rankings is made up of $g_1$ as the first group and a cyclic sweep of the remaining items to fill the remaining positions. Therefore the optimal permutation must be among the sets created by considering each of the $2k + 1$ groups as the first group in the rankings. Let $R_{i,j}$ represent the ranking which has group $g_i$ as the first group and the remaining groups present as a cyclic sweep from $g_j$. Consider the case of $R_{1,k}$. Let $x_i$ represent the number of items in group $g_i$. If $R_{1,k}$ is a better ranking than $R_{k,k+1}$, it implies that

$$\sum_{i=k}^{n+1} x_i > \sum_{i=n+2}^{2n+1} x_i \tag{1}$$

by considering the shift of $g_1$ in the two rankings. The difference in the number of upsets between $R_{1,2}$ and $R_{1,k}$ is given by

$$x_2(-x_k - x_{k+1} \dots - x_{n+2} + x_{n+3} \dots x_{2n+1}) + x_3(-x_k - x_{k+1} \dots - x_{n+3} + x_{n+4} \dots x_{2n+1}) \dots$$

Using Equation 1, it can be seen that each of the above terms are negative for any $j \leq n + 1$, making $R_{1,2}$ the better ranking. Any $j > n + 1$ cannot be considered as an optimal ranking since the first group as per the ranking must precede the second(otherwise switching them would decrease the upsets). Since either $R_{1,2}$ or $R_{k,k+1}$(both counterclockwise orderings) is better than $R_{1,k}$ whenever $k \leq n + 1$($R_{1,k}$ cannot be the optimal ranking when $k > n + 1$), and since this can be generalised for any $R_{i,j}$, it is shown that one of the counterclockwise orderings of the items is the optimal ranking. $\square$

We note that the above proof also appeared in **?**. However, the insights via flip classes were not present there.

A.0.1  FINDING FORBIDDEN CONFIGURATIONS

Given a tournament and a representation dimension, there is no known method to check whether the tournament can be represented by vectors in the given dimension. The forbidden configurations presented above were found by carefully creating an exhaustive set of cases, and showing each one causes a contradiction. The techniques used are presented below.

**Proof of Theorem 8**

*Proof.* Consider the representation of tournaments as given in Subsection 3. Since adding minor noise to each $h_i$ will not change tournament, we can deal only with cases in which any size $4$ subset of $h_1, h_2, ...h_1 2$ consists of linearly independent vectors. By using this property, and without loss of generality,

$$c_1 h_1 + c_2 h_2 + c_3 h_3 + c_4 h_4 = h_5 \tag{2}$$

We can construct cases based on the signs of the coefficients($c_i$) in the above equation. There are $2^4 = 16$ sign patterns/cases for any equation. These cases are filtered by multiplying the entire expression with expressions of the form $\mathbf{A}^{\mathrm{rot}} h_i$.

$$c_1 h_1 \mathbf{A}^{\mathrm{rot}} h_i + c_2 h_2 \mathbf{A}^{\mathrm{rot}} h_i + c_3 h_3 \mathbf{A}^{\mathrm{rot}} h_i + c_4 h_4 \mathbf{A}^{\mathrm{rot}} h_i = h_5 \mathbf{A}^{\mathrm{rot}} h_i \tag{3}$$

In the above equation, the signs of all expressions of the form $h_i \mathbf{A}^{\mathrm{rot}} h_j$ is known from the tournament configuration. Therefore simply comparing the sign of the LHS and RHS for all possible values of $i$ rules out many cases.

We now use multiple equations together in a bid to further filter the remaining cases. Consider the following equations without loss of generality.

$$c_1 h_1 + c_2 h_2 + c_3 h_3 + c_4 h_4 = h_5 \tag{4}$$

$$b_1 h_2 + b_2 h_3 + b_3 h_4 + b_4 h_5 = h_6 \tag{5}$$

$$a_1 h_1 + a_2 h_2 + a_3 h_3 + a_4 h_4 = h_6 \tag{6}$$

$h_5$ can be eliminated from the first two equations, leaving two expressions of $h_6$ in terms of $h_1, \ldots h_4$. The two sets of coefficients of $h_1, \ldots h_4$ can be equated, and sign based arguments can eliminate a few more cases. Also, a set of 3 tuples can be constructed, with each item representing possible sign patterns for the 3 equations. Note that this set of 3 tuples does not contain entries which cannot apply simultaneously on the 3 equations.

The above elimination step can be considered as a filtration procedure given a size 6 tuple $(h_1, h_2, h_3, h_4, h_5, h_6)$ by using 3 equations. We can perform a similar filtration step given any size 6 tuple as well. Let the equations corresponding to $(h_1, h_2, h_3, h_4, h_5, h_6)$ be $E_{1,1}, E_{1,2}, E_{1,3}$. Similarly, consider the tuples $(h_1, h_2, h_3, h_4, h_5, h_7), (h_2, h_3, h_4, h_5, h_6, h_7), (h_1, h_2, h_3, h_4, h_6, h_7)$ and their corresponding equations represented by $E_{i,j}$ where $i$ is the index of the tuple it corresponds to and $j$ is the index of the equation. The above filtration process can be performed on all the tuples, following which an additional filtering step can be performed by using

- $E_{1,1} \equiv E_{2,1}$

- $E_{1,2} \equiv E_{3,1}$

- $E_{1,3} \equiv E_{4,1}$

- $E_{2,2} \equiv E_{3,3}$

- $E_{2,3} \equiv E_{4,2}$

where the equivalence relation represents that the 2 equations are identical. Since identical equations must have identical sign patterns, the sign patterns not present in both sets of possible sign patterns can be filtered out. For the 11-DRT cone, this leaves us with null set, proving that it is a forbidden configuration. $\square$

**Proof of Theorem 9**

*Proof.* The result follows the arguments in Forster & Simon (2006). We note that the arguments in Forster & Simon (2006) work only for non-zero sign matrices. By overloading $\mathbf{T}$ to denote the signed adjacency matrix of the corresponding tournament, we can consider the non-zero sign matrix $\mathbf{G} = \mathbf{T} + \text{diag}(\mathbf{b})$ where $\mathbf{b} \in \{1, -1\}^n$. By Gershgorin's circle theorem and exploting the fact that any two rows of a doubly regular tournament are orthogonal, one can show that $\rho_i(\mathbf{G})^2 \leq \rho_i(\mathbf{T})^2 + 2n - 2$ where $\rho_i$ denotes the $i$-th largest singular value of the corresponding matrix. It then follows from Forster & Simon (2006) that if $\mathbf{G}$ has an representation in $\text{dim}$ dimensions, then it must be the case that

$$\text{dim} \sum_{i=1}^{\text{dim}} (\rho_i(\mathbf{G}))^2 \geq n^2.$$

Noting that for a doubly regular cone $\mathbf{T}$ on $n$ nodes, $\rho_i(\mathbf{T}).^2 = n - 1 \; \forall i$, we get

$$\text{dim}^2 \cdot 3(n-1) \geq n^2 => \text{dim} \geq \sqrt{\frac{n}{3}} \qquad (7)$$

Thus to get a matrix $\mathbf{M}$ such that $sign(\mathbf{M}) = \mathbf{G}$, one needs at least $\sqrt{\frac{n}{3}}$ dimensions i.e., $rank(\mathbf{M}) \geq \sqrt{\frac{n}{3}}$.

Now we show that this also is a lower bound for representing $\mathbf{T}$. To see this, assume for the sake of contradiction that $\mathbf{T}$ has a representation in $d$ dimension where $d < \sqrt{\frac{n}{3}}$. Then, there exists $\mathbf{H} \in \mathbb{R}^{d \times n}$ such that $\mathbf{T} = \mathbf{T}[\mathbf{H}]$. The diagonal entries of $\mathbf{T}[\mathbf{H}]$ are zero. Now introduce a small enough perturbation $E$ to $\mathbf{H}$ and consider the matrix $(\mathbf{H} + E)^T \mathbf{A}^{\text{rot}} \mathbf{H}$. The entries of the perturbation matrix $E \in \mathbb{R}^{d \times n}$ can be chosen to be small enough such that the sign of the off-diagonal entries of $\mathbf{H}^T \mathbf{A}^{\text{rot}} \mathbf{H}$ is same as that of $(\mathbf{H} + E)^T \mathbf{A}^{\text{rot}} \mathbf{H}$. However, the diagonal entries of $(\mathbf{H} + E)^T \mathbf{A}^{\text{rot}} \mathbf{H}$ can get an arbitrary sign pattern, say $\mathbf{b}$. Let $\mathbf{G}$ be the sign pattern matrix corresponding to $(\mathbf{H} + E)^T \mathbf{A}^{\text{rot}} \mathbf{H}$. By definition, $\mathbf{G}$ has a representation in dimension $d < \sqrt{\frac{n}{3}}$. But this contradicts inequality (7). Thus, it must be the case that $\mathbf{T}$ has a representation in dimension at least $\sqrt{\frac{n}{3}}$.

Finally, setting $n = 12d^2$ to the number of nodes in the doubly regular cone as given in the Theorem, we get that at least $2d$ dimensions are needed to embed such a tournament. The result follows. $\square$

**Proof of Theorem 10**

*Proof.* The proof is the same arguments as the first part of the proof of Theorem 9. $\square$

**Proof of Theorem 11**

*Proof.* Given a tournament $\mathbf{T}$, we first show that we can start with an arbitrary transitive tournament and add enough rank 2 *corrections* to obtain a representation for $\mathbf{T}$. Every addition of a rank 2 matrix will increase the representation dimension by at most 2. The result will follow then by noting that for the choice of the transitive tournament which minimizes the number of corrections needed, one needs at most $2(\mu(\mathbf{T}) + 1)$ representation dimension.

Let $\mathbf{T}'$ be an arbitrary transitive tournament which has a 2 dimensional representation and let $\mathbf{M} \in \mathbb{R}^{n \times n}$ be the associated skew symmetric matrix that represents $\mathbf{T}'$. W.l.o.g, assume that $M_{ij} > 0$ if and only if $i < j$. Define $E_k(\mathbf{T}) = \{i : i < k, \; i <_{\mathbf{T}} k\}$. By definition, the feedback arc set $E^\sigma(\mathbf{T}) = \cup_{k=1}^n (k, E_k(\mathbf{T}))$ where $\sigma = [1, \ldots, n]$ is the topological order corresponding to the transitive tournament $\mathbf{T}'$. We start with $k = n$ and *correct* the feedback arc errors arising from $E_k$ iteratively. Define $\Delta_n = \sqrt{\max_{i<n}(M_{in} + \epsilon)}$ for some small enough $\epsilon > 0$. Let $\mathbf{u} \in \mathbb{R}^n$ be such that $u_i = \Delta \; \forall i \in E_n$ and $u_i = 0$ otherwise. Let $\mathbf{v} \in \mathbb{R}^n$ be such that $v_n = -\Delta, v_i = 0 \; \forall i \neq n$. It is easy to verify that $\mathbf{M} + \mathbf{u}\mathbf{v}^T - \mathbf{v}\mathbf{u}^T$ represents a tournament that has the feedback arc set $\cup_{k=1}^{n-1}(k, E_k(\mathbf{T}))$ i.e., the errors in $E_n$ have been corrected. The cost of correcting the error is adding a rank 2 skew

symmetric matrix which increases the representation dimension by at most $2$. One can repeat the same procedure for $n-1, n-2, \ldots$ until all errors are corrected.

The upper bound in the theorem follows noting that the above argument works for any transitive tournament $\mathbf{T}'$ and so we can start with the one which has the least number of nodes involved in the feedback arc set to minimize the number of extra dimensions needed to represent $\mathbf{T}$. ☐

**Proof of Theorem 12**

*Proof.* From Theorem 11, $\mathbf{T}$ can be represented using at most $2(\mu(\mathbf{T})+1)$ dimensions. By construction any representation of $\mathbf{T}$ must also represent $\mathbf{G}$. The result follows. ☐

### A.0.2 ADDITIONAL FILTRATION FOR VERIFYING 10 NODE FORBIDDEN CONFIGURATION

$$\mathbf{T} = \begin{pmatrix} 0 & 1 & 1 & 1 & -1 & -1 & -1 & -1 & -1 & 1 \\ -1 & 0 & 1 & -1 & 1 & 1 & -1 & -1 & 1 & -1 \\ -1 & -1 & 0 & 1 & 1 & -1 & 1 & -1 & -1 & -1 \\ -1 & 1 & -1 & 0 & -1 & 1 & 1 & -1 & -1 & -1 \\ 1 & -1 & -1 & 1 & 0 & 1 & -1 & -1 & 1 & -1 \\ 1 & -1 & 1 & -1 & -1 & 0 & 1 & -1 & 1 & -1 \\ 1 & 1 & -1 & -1 & 1 & -1 & 0 & -1 & 1 & -1 \\ 1 & 1 & 1 & 1 & 1 & 1 & 1 & 0 & 1 & -1 \\ 1 & -1 & 1 & 1 & -1 & -1 & -1 & -1 & 0 & -1 \\ -1 & 1 & 1 & 1 & 1 & 1 & 1 & 1 & 1 & 0 \end{pmatrix}$$

Due to the lesser number of constraints present when dealing with 10 nodes instead of 12, an additional filtration step is required. The entire procedure described above can be considered as an elimination procedure given a tuple $T_1 = (h_1, \ldots h_7)$. Also consider that in the final filtration above, a set of size 4 tuples is created. These size 4 tuples will represent the possible sign patterns for $E_{1,1}, E_{1,2}, E_{1,3}, E_{2,2}$. If a similar procedure is carried out on $T_2 = (h_2, \ldots h_8)$, another set of 4 tuples will be generated. The filtration in this step is based on the fact that the equations $E_{1,3}, E2, 2$ corresponding to $T_1$ are the same as $E_{1,1}, E_{1,2}$ for $T_2$. The sign patterns which are not present in both sets of tuples are filtered out. This type of relationship can be obtained between $T_2$ and $T_3 = (h_3, \ldots h_8, h_0)$ as well, and in general between any $T_i$ and $T_j$ such that $T_j$ can be obtained from $T_i$ by removing the first entry and adding an entry to the end. Note that the tuples must have unique entries. Therefore a 'circular elimination' procedure can be followed, where the $T_i$ tuples considered are size 7 subsets of the circular permutations of $h_1, \ldots h_8$. By following this procedure, all the sign patterns remaining can be eliminated. Using size 7 circular permutations of $h_1, \ldots h_1 0$ in the last step does not eliminate all the cases, we believe this is due to nodes 1 to 8 forming a 7-DRT cone, which leads to many eliminations. The code for the filtration can be found *here*.

