# OpenReview forum: "A Theory of Tournament Representations"
_ICLR.cc/2022/Conference — ICLR 2022 Poster_

### Official Review · Reviewer_YSHR · 2021-11-02

**Correctness:** 3
**Technical Novelty And Significance:** 3
**Empirical Novelty And Significance:** 2
**Recommendation:** 5
**Confidence:** 4

**Main Review:**

Strengths:
- the paper introduces a novel theory of tournament representations, leveraging their specific properties that separate them from sign matrices
- the connection between representations, flip classes and cones is made clear and nicely leveraged
- a complete theory of rank 2 tournaments is obtained, which corresponds to the intuitively "simplest" tournaments

Weaknesses:
- motivation: the applications of such tournaments representations, which are much more restrictive than general sign matrices, are never mentioned; the ones in Alon et al. do not apply to this setting
- interpretability: much attention is given to vector representations $H$ where $H^\top A^{\mathrm{rot}} H = M$, but no interpretation of those $h_i$ is given, in terms of how the positions of those vectors relate to the tournament structure. The proof of Lemma 3 would also be easier to understand if the connection between rank-2 representation and angles was made clearer.
- Theorem 6 is neat, but its connection to the rest of your work is tenuous: the proof only uses the fact that $T$ is locally transitive, and I fail to see any situation where showing that a tournament has rank 2 is easier than showing it's locally transitive
- the quantity $\mu(T)$ involves minimizing the feedback vertex set under all $2^n$ possible flips, hence stacking an exponential problem on top of an NP-hard one, known to be very hard to approximate. Maybe this would benefit from an example where the lower bound given by $\mu(T)$ is better than the ones in Alon et al.

**Summary Of The Paper:**

This paper studies the theory of tournament representations, i.e. low-rank matrices $M$ whose sign agrees with the sign matrix of a tournament $T$.

The authors show several properties of such representations, reducing the study to so called $R$-cones, i.e. tournaments where one vertex beats all others. They also characterize completely rank 2 tournaments, and provide a forbiden class for rank $d$ tournaments.

Finally, they provide an upper bound on the minimum dimension of a presentation of any tournament $T$, in the for of a bound involving minimum feedback arc sets of $T$, and show how this can be extended to sign matrices.

**Summary Of The Review:**

This paper introduces a specification of sign matrix representation to tournaments, with some interesting ideas, but the significance and improvements over Alon et al. are not made clear enough to warrant acceptance.

---

> ### Author Response · Authors · 2021-11-09
> **Answering Review Questions**
>
> Thanks for your detailed review.
>
> “motivation: the applications of such tournaments representations, which are much more restrictive than general sign matrices, are never mentioned; the ones in Alon et al. do not apply to this setting”
>
> 	- The motivation of Alon et.al arise from studying sign patterns of binary hypothesis classes (in a Machine Learning context), communication complexity etc. This is NOT our motivation. Our motivation to study tournament representations arise from problems related to pairwise comparisons - ranking, match prediction, etc. Here, skew symmetric sign pattern matrices arise naturally. This additional skew symmetry of these sign matrices help us exploit the intricate structure which was not possible for earlier works in Alon et.al etc where the problem of interest was different.  This is not an extension of the work of Alon et.al. We just point out connections to their work but do not build on them.
>
> interpretability: …The proof of Lemma 3 would also be easier to understand if the connection between rank-2 representation and angles was made clearer.
>
> 	- There are several ways to prove Lemma 3. The geometric interpretation is one simple way, but not necessarily the only way. Consider three nodes i and j with 2-dimensional representations h_i =[h_i1 h_i2]’ and h_j =[H_j1 h_j2]’ and h_k = [h_k1 h_k2]. Let i  beat j. Then by definition h_i’A^{rot}h_j > 0. In two dimension, this quantity is simply the cross product between vectors h_i and h_j i.e., h_i’A^{rot}h_j = h_i1*h_j2 - h_j1*h_i2.  Thus the quantity is positive if and only if the counter clockwise angle between the vectors is at most 180 degrees. This observation can be used to prove Lemma 3 as follows: If i > j> k > i, then the counter clockwise angle between i and j is at most 180 degrees and the same holds for j and k and k and i as well. If these vectors did not positively span the entire space, then there must be a line (hyperplane) passing through the origin such that h_i, h_j and h_k all lie on one side of the line. But this means that the counter clockwise direction between at least one pair of nodes must be greater than 180 degrees which is a contradiction.
>
>     Non-geometric Approach: The other non-geometric way to prove this statement would be using the theory of oriented matroids. The sign patterns formed by any hyperplane arrangements is an oriented matroid. Using axioms of oriented matroid, one can show that if a 3-cyle cone was possible in two dimensions, then there are enough hyperplanes passing through the origin to “shatter” 3 points in the Vapnik-Chevronenkis sense. But this is a contradiction to the well known fact that the VC dimension of hyperplane classifiers passing through origin is 2 in R^2.
>
> “I fail to see any situation where showing that a tournament has rank 2 is easier than showing it's locally transitive” -
>
>
>      The situation we are interested are not in the ease of “showing” a tournament is rank 2 or locally transitive but in the ease of “modelling”. To see this, consider a simple situation of modelling sports tournaments such as Tennis. Here, one can model the players (nodes) using  2-dimensions where the dimensions corresponds to their “offense” and “defense” strengths respectively. When two players compete, the advantage of the offense strength of player 1 w.r.t the defense of player 2 and vice versa determine the outcome of the match. This is precisely captured by a rank 2 tournament where the learning problem would be to infer these latent offense/defense strength of each player from pairwise competitions. Theorem 6 says that such a model would immediately lead to locally transitive tournaments among the players. This structural characterisation now gives insights to the modeller if 2 dimensions are enough to model the players or not.
>
> “Maybe this would benefit from an example where the lower bound given by \mu(T) is better than the ones in Alon et al” - We first clarify that \mu(T) is an upper bound and not a lower bound on the representation dimension.
>
>      This is a good point and we argue below that the gain using \mu(T) could be significant. The simplest example one can consider is a locally transitive tournament on N nodes. Viewing this as a sign pattern matrix, it is easy to argue that the VC dimension of such a matrix is 2. Theorem 5 in Alon et al shows that the sign rank is upper bounded by O(sqrt(N)) when VC dimension is 2. However, \mu(T) is exactly 0 (and so the upper bound of 2 \mu(T) + 1 is exactly 2) for any locally transitive tournament which is independent of the number of nodes N. The important reason for this significantly improved bound using \mu(T)  is because we consider only skew symmetric sign pattern matrices while Alon et.al look at worst case (w.r.t representation dimension/sign rank)  sign pattern matrices for a given VC dimension.

---

> > ### Comment · Reviewer_YSHR · 2021-11-16
> > **Review reassessment**
> >
> > Thank you for your detailed reply.
> >
> > In light of your other answers to my and other reviewers' comments, my opinion has changed on the significance and novelty of the results; I have updated my initial review accordingly.
> >
> > On the other hand, the fact that your paper needed this much clarification in the comments is indicative of a broader clarity issue, that needs to be addressed. Hence, my assessment stays below the acceptance threshold, but only marginally.

---

> > > ### Author Response · Authors · 2021-11-17
> > > **Thanks**
> > >
> > > Thank you for reconsidering your opinion about our work. We are glad that the response helped clarify the concerns about the significance/novelty. We will submit an updated draft of the paper shortly incorporating all the valuable reviewer suggestions.

---

### Official Review · Reviewer_AhwX · 2021-11-02

**Correctness:** 4
**Technical Novelty And Significance:** 2
**Empirical Novelty And Significance:** Not applicable
**Recommendation:** 5
**Confidence:** 3

**Main Review:**

Tournaments are important tools in sports modeling, social preference, etc. A tournament on n nodes is the complete directed graph where every pair of nodes has a directed edge pointing from one vertex to the other. One can represent a tournament as the sign matrix of a rank d matrix. In this work, the authors develop a novel theory for understanding tournament representation.

Firstly, the authors characterize the structure of rank d tournaments using a list of forbidden configurations. These forbidden configurations are formed by the union of so-called flip classes. For rank 2 tournaments, the authors use this framework to describe precisely the forbidden configurations and show that they are actually equivalent to locally transitive tournaments, a class of tournaments closely related to transitive tournaments. The authors also show one forbidden flip class for rank 4 tournaments and also a weaker forbidden flip class for the general rank d case.

Secondly, the authors show both lower and upper bound on the minimum dimension required for tournament representation on n nodes. The lower bound of order sqrt(n) is proved by a variation of the celebrated work of Forster and Simon for sign matrices. The upper bound is given in terms of a parameter named the Flip Feedback Node set of a Tournament introduced by the authors. Such an upper bound also gives new bounds for sign ranks of matrices.

The results of this paper are overall nice, but I find some of results in the first part slightly weak. Especially for rank-4 and rand-d tournaments the forbidden flip classes are still not totally clear. Also, it is unclear to me if the minimum feedback arc set problem can be solved efficiently for constant rank representable tournaments. Is this problem efficiently solvable when the representation matrix is given? In fact, more discussions about applications of low-rank representations or forbidden flip classes would be appreciated. As a potential direction for improvement or future direction, one can probably also try studying the representation dimension of random tournaments.

Detailed comments:
Page 2: “We exhibit a lower bound of O(\sqrt(n))”
Technically you should use \Omega() for lower bound instead of big-O
Page 4, Definition 3: I think in the first item you should have “for all j \neq i^*” instead of i


**Summary Of The Paper:**

This paper provides fundamental theories of tournament representations. The authors study two main questions. First they characterize the class of tournaments that can be represented in d dimensions. Second they give lower and upper bounds on the minimum dimension needed to represent a tournament on n nodes.


**Summary Of The Review:**

Overall this is a nice paper though I think there is still room for improvements for some parts. Some results are slightly weak and I don’t really see any application for most results. Perhaps the authors should discuss more about the application parts.

---

> ### Author Response · Authors · 2021-11-09
> **Answering Questions Raised by the Reviewer**
>
>
> Thanks for your detailed review. Please find answers to the questions raised in the review.
>
> “Especially for rank-4 and rand-d tournaments the forbidden flip classes are still not totally clear.”
>
> 	- This appears to be a non-trivial problem. From our synthetic experiments, we observe that the “only” flip class that could be forbidden on 8 nodes is the one that contains the 7-doubly regular cone. In particular, we were able to produce examples of representations for all other flip classes on 8 nodes. However, this does not imply that the ones where we could not produce a forbidden configuration is in fact forbidden. Unfortunately, it seems tricky to prove this and we don’t have a way to show this at this point.
>
> “Also, it is unclear to me if the minimum feedback arc set problem can be solved efficiently for constant rank representable tournaments.”
>
> 	- In this work,  we answer this in the affirmative only for rank 2. For a constant rank, this question can be answered only when we understand the precise forbidden configurations. It would be nice to have a parametrised complexity result for solving the feedback arc set problem where the parameter of interest is the rank. This is part of future work. We note that solving the feedback arc set to obtain a ranking is only one of the problems of interest. In general, one may be interested in learning to predict edge directions from a subset of pairwise comparisons where one would still need to understand the space of forbidden configurations.
>
> “Is this problem efficiently solvable when the representation matrix is given?”
>
> 	- Again this is true for the rank 2 case. In general, knowing the representation means knowing the associated tournament. If one can show that the associated tournament comes from an efficiently solvable flip class, then yes.
>
> In fact, more discussions about applications of low-rank representations or forbidden flip classes would be appreciated.
>
> 	- Low rank representations have been studied earlier. See for example the blade-chest model of Suo Chen et al., LRPR model of Rajkumar et al. We have mentioned this in the related work section.
>
> As a potential direction for improvement or future direction, one can probably also try studying the representation dimension of random tournaments.
>
> 	- This is a good direction. In general, the lower bound for the representation dimension of random tournaments will depend on the singular values of random tournaments. However, we don’t expect the representation dimension to be of independent of n, the number of nodes. A Doubly regular tournament is "like" a random tournament (the associated Hadamard Tournament has RIP properties similar to random tournaments). On the other hand, the most interesting real-world tournaments  might be characterized by constant sized node representations and hence may be structurally much more constrained  than random tournaments.

---

### Official Review · Reviewer_UVYb · 2021-11-03

**Correctness:** 3
**Technical Novelty And Significance:** 3
**Empirical Novelty And Significance:** Not applicable
**Recommendation:** 6
**Confidence:** 3

**Main Review:**

The goal of connecting dimensional characterization and structural characterization is important and the connection to flip classes very interesting. The results seem to be sound, although I had non-insignificant troubles trying to verify them due to the writing style (see next paragraph on weaknesses)

I think that the paper needs a thorough stylistic rehandling and it is not ready to be published in its present form. The preliminary section does not help to orientate in the dense notation and terminology employed trhoughout the paper:
Concrete examples:
- the definition of tournament is not precise (it is the underlying undirected graph to be complete)
- the notation H={h1,..,hn} \in R^d is very confusing. You mean each h_i \in R^d .
- the description of representation and canonical form of skew-symmetric matrices is confusing also because of the previous point
- btw, in a preliminary section, also the definition of skew symmetric matrix could find its place, since you are defining other basic concepts too
- proof of Lemma 2: "H realizes R" has never been defined before
- line after Definition 4: "3-doubly regular tournament" is not defined (what is 3); this notation is also used later (e.g., 11-doubly... in Thm 8)
- Thm 7: the sign of T is not defined.
- ...

In conclusion, I think the paper contains some interesting ideas and results however I am not fully satisfied with the clarity of the arguments. I have read and appreciated some of the authors updates and clarifications, following their reply to my questions. Admittedly, the paper needed significant explanations from the authors. I have updated my score which is now less negative but still only marginally above the acceptance.

**Summary Of The Paper:**

The paper studies the relationship between dimensional representation of tournament and their structural characterization. In particular, a relationship is established between rank d tournament and their forbidden configurations in terms of flip classes, introduced by Fisher&Ryan(1995) as a way to partion the set of tournaments of a given order. In addition, the problem of bounding the minimum possible dimension of a representation of a tournament is also investigate and lower and upper bounds are given.

**Summary Of The Review:**

It was not possible for me to fully assess the correctness of the claims. The ideas and claims are of interest for the area, however the present form of the presentation is below the bar of acceptance, in my opinion.

[Following the authors' reply] I have read and appreciated some of the authors updates and clarifications, following their reply to my questions. Admittedly, the paper needed significant explanations from the authors. I have updated my score which is now less negative but still only marginally above the acceptance.

---

> ### Author Response · Authors · 2021-11-09
> **Clarifying the Points Raised by the Reviewer**
>
> Thanks for your detailed  review. We clairfy the points raised below:
>
> “the definition of tournament is not precise (it is the underlying undirected graph to be complete)”
>
> 	A tournament is a complete directed graph. This is a standard graph theoretic definition of a tournament.
>
> “the notation H={h1,..,hn} \in R^d is very confusing. You mean each h_i \in R^d.”
>
> 	Yes, every h_i \in R^d.  In other words, if one arranges these vectors as columns of a matrix i.e. if H = [h_1|h_2|…|h_n], then H is a R^(d * n) matrix.
>
> “the description of representation and canonical form of skew-symmetric matrices is confusing also because of the previous point”
>
> 	- The canonical form of the skew-symmetric matrix is a standard well known result: Any non-degenrate skew symmetric bilinear form can be brought to the A^rot form using a basis change. See for instance https://en.wikipedia.org/wiki/Symplectic_matrix
>
> “btw, in a preliminary section, also the definition of skew symmetric matrix could find its place, since you are defining other basic concepts too”
>
> 	- A skew symmetric matrix M is one where M = -M^{T}.
>
> “proof of Lemma 2: "H realizes R" has never been defined before”
>
> 	- H realizes R if the tournament induced by H is R
>
> “line after Definition 4: "3-doubly regular tournament" is not defined (what is 3); this notation is also used later (e.g., 11-doubly... in Thm 8)”
>
> 	- 3 is the number of nodes of the doubly regular tournament considered.
>
>
> “Thm 7: the sign of T is not defined”
>
> 	- sign of (i,j) in T is the direction of orientation of the edge between i and j. When i and j are compared, the outcome is “i beats j” if the edge is oriented from i to j in T and “j beats i” if the edge is oriented from j to i.
>
> We hope that with these clarifications, the reviewer will be able to verify the correctness of the claims. Happy to provide any further clarifications needed.

---

### Official Review · Reviewer_MMVk · 2021-11-03

**Correctness:** 4
**Technical Novelty And Significance:** 3
**Empirical Novelty And Significance:** 1
**Recommendation:** 8
**Confidence:** 4

**Main Review:**

I think this paper has a solid theoretical foundation of an interesting problem on tournaments.

My main concern is if this conference is the right venue since I fail to see the connection to "learning." The authors briefly address applications in Section 7.2, but I think this section should be expanded. Some of the proofs can be moved to the appendices to make space if need be.

Minor errors:

“Recent works have noted that parametric models which assume d dimensional node representations can effectively model intransitive tournaments” are there any citations for this statement? You might want to list them in the introduction where something similar is also mentioned.


No “.” after 2) point at the end of Page 1.

In the definition of skew symmetric, it would be easy to add A^T = - A

The name of the Lemma is Farkas’ Lemma, and not Farka’s lemma.

In the citation of Charbit et al, the “np” should be capitalized.

**Summary Of The Paper:**

A tournament is made by choosing a direction for each of the edges in a complete graph. A tournament can be induced of by skew symmetric matrices M where entries M_{ij} > 0 if and only if (i,j) is an edge. A tournament on n edges can be represented by a set of d-dimensional vectores {h_1, … , h_n} if (h_j)^T A h_i is not zero iff (i,j) is an edge (and A is an appropriate matrix).

The authors address two questions:

1) What structurally characterizes the class of tournaments that can be represented in d dimensions?

2) Given a tournament T on n nodes, what is the minimum dimension d needed to represent it?.

The first question is answered by considering structures the authors called forbidden. The authors provide a characterization of these forbidden structures as a union of certain equivalence classes.

They answer the latter question in part by providing bounds on what said dimension d should be.



**Summary Of The Review:**

Overall, I think the paper should be accepted provided that the authors make more of an effort to relate their results to learning.

---

> ### Author Response · Authors · 2021-11-09
> **Discussion about the Learning Problem**
>
> Thanks for appreciating the work.
>
> “Overall, I think the paper should be accepted provided that the authors make more of an effort to relate their results to learning”
>
>     The learning problem is as follows:
>     Consider a learning to rank problem from pairwise comparisons. Here, a set of n items need to be ranked from a subset of pairwise comparisons among them. Every pair that is chosen for comparison can be compared multiple times and every time items i and j are compared, i is preferred over j with probability P_ij. A common and popular model to capture these probabilities is the Bradley-Terry-Luce (BTL) model where P_ij = s_i/(s_i + s_j) for some score vector s. Note that the model is completely specified by the score vector s \in R^n which in turn completely determines the probability preference matrix P.  The learning problem is to learn these unknown score vector s \in R^n from pairwise comparisons. Once the score vector is learnt, a ranking can be obtained by sorting these scores.
>
>     The above BTL mode model is an example of a rank-2 model in the sense that the probability preference matrix P results in a rank 2 matrix under the log-odds skew symmetric transformation. Indeed, if we deine  M_ij = log(P_ij/P_ji). then equialently M_ij = log(s_i) - log(s_j). It is easy to show that M is a rank 2 skew symmetric matrix. However, the major disadvantage of the BTL model is that it can capture only transitive preferences i..e, P_ij > 0.5 and P_jk > 0.5 => P_ki > 0.5.  In real world situations, intransitivity is very common. To achieve intransitivity, the simplest way would be to start with a general rank 2 skew symmetric matrix M and consider the probability matrix that determines the preference probabilities as P_{ij} = 1/(1 + exp(-M_ij}). Here, the learning problem would be to estimate the two score vectors or equivalently the two dimensional representation for each item. Previous studies show that this can be learnt using Matrix completion based approaches (See for example Low Rank Pairwise Ranking model of Rajkumar et.al) or maximum likelihood based approaches (Suo Chen et.al). However, what was not known earlier is the structure of tournaments that can be captured using such low rank restrictions. This is what we address in this work.
>
> We would be happy to expand this discussion in the final version.

---

### Author Response · Authors · 2021-11-20
**Uploaded New Version**

A new version incorporating reviewer comments has been added. The main changes include moving the proofs to the appendix and adding detailed discussion on points raised by the reviewers. We hope the updated version addresses reviewer concerns. Happy to discuss further if there are more questions/comments.

---

### Decision · Program_Chairs · 2022-01-20

**Decision:**

Accept (Poster)

**Comment:**

The paper takes a creative step in the theory of tournaments, and it seems plausible that this could lead to interesting follow-ups. The reviewers made many excellent comments and I highly encourage the authors to take ALL of them into account in the revision, it will make the paper much stronger.